# Vitamin C Cytotoxicity and Its Effects in Redox Homeostasis and Energetic Metabolism in Papillary Thyroid Carcinoma Cell Lines

**DOI:** 10.3390/antiox10050809

**Published:** 2021-05-20

**Authors:** Laura Tronci, Gabriele Serreli, Cristina Piras, Daniela Virginia Frau, Tinuccia Dettori, Monica Deiana, Federica Murgia, Maria Laura Santoru, Martina Spada, Vera Piera Leoni, Julian Leether Griffin, Roberta Vanni, Luigi Atzori, Paola Caria

**Affiliations:** 1Department of Biomedical Sciences, University of Cagliari, Cittadella Universitaria, SS 554, km 4.5, 09042 Monserrato, Italy; lauratronci90@gmail.com (L.T.); cristina.piras@unica.it (C.P.); dvfrau@unica.it (D.V.F.); dettorit@unica.it (T.D.); mdeiana@unica.it (M.D.); federica.murgia@unica.it (F.M.); marialaurasantoru@gmail.com (M.L.S.); martina.spada@unica.it (M.S.); vleoni@unica.it (V.P.L.); vanni@unica.it (R.V.); latzori@unica.it (L.A.); paola.caria@unica.it (P.C.); 2Department of Biochemistry & Cambridge Systems Biology Centre, University of Cambridge, Cambridge CB1 9NL, UK; julian.griffin@imperial.ac.uk

**Keywords:** vitamin C, PTC cells, ROS, cell metabolism, TCA cycle, antioxidants, glycolysis, anticancer effects

## Abstract

High-dose of vitamin C (L-ascorbic acid, ascorbate) exhibits anti-tumoral effects, primarily mediated by pro-oxidant mechanisms. This cytotoxic effect is thought to affect the reciprocal crosstalk between redox balance and cell metabolism in different cancer types. Vitamin C also inhibits the growth of papillary thyroid carcinoma (PTC) cells, although the metabolic and redox effects remain to be fully understood. To shed light on these aspects, PTC-derived cell lines harboring the most common genetic alterations characterizing this tumor were used. Cell viability, apoptosis, and the metabolome were explored by 3-(4,5-dimethylthiazol-2-yl)-2,5-diphenyltetrazolium bromide test (MTT), flow cytometry, and UHPLC/MS. Changes were observed in redox homeostasis, with increased reactive oxygen species (ROS) level and perturbation in antioxidants and electron carriers, leading to cell death by both apoptosis and necrosis. The oxidative stress contributed to the metabolic alterations in both glycolysis and TCA cycle. Our results confirm the pro-oxidant effect of vitamin C as relevant in triggering the cytotoxicity in PTC cells and suggest that inhibition of glycolysis and alteration of TCA cycle via NAD^+^ depletion can play an important role in this mechanism of PTC cancer cell death.

## 1. Introduction

Several in vitro and in vivo studies have shown that a high dose of vitamin C exhibits antitumor effects [1,2,3], and this cytotoxicity is mediated by the accumulation of hydrogen peroxide (H_2_O_2_), resulting in the depletion of intracellular antioxidants [4,5,6,7]. Particularly, the accumulation of H_2_O_2_ leads to the conversion of the reduced glutathione (GSH) into the oxidized form (GSSG), resulting in oxidative stress and cell damage [6]. It is becoming clear that redox perturbations and altered energy metabolism are hallmarks of cancer, playing a fundamental role in the development, progression, and survival of cancer cells [8]. Previous studies showed the effect of vitamin C on the viability of breast and colon cancer cells, by inhibiting energetic processes and consequently ATP production [4,9]. The selective action of high levels of vitamin C on cultured cancer cells harboring *KRAS* or *BRAF* mutations has been demonstrated for the first time in colon cancer, which commonly harbors these genetic mutations [9]. This observation provided a mechanistic rationale for exploring the therapeutic use of vitamin C in treating cancer cells with these mutations. Interestingly, *KRAS* or *BRAF* mutations, as well as *RET/PTC* rearrangements, characterize the vast majority of papillary thyroid carcinoma (PTC) [10]. These molecular changes activate the MAPK/ERK and PI3/AKT pathways, which are affected in thyroid cancer cells exposed to a high dose of vitamin C [11]. Nevertheless, the exact role of vitamin C in perturbing metabolism in PTC cancer cells is still poorly understood. PTC accounts for 80–85% of differentiated thyroid cancer and, in general, it has a good prognosis and a relatively low aggressiveness [12]. However, disease progression and patient death may occur due to poor response or resistance to standard treatment. In the present research, the anti-tumoral mechanism of vitamin C in a PTC in vitro model has been investigated by exploring oxidative effects and metabolic alterations. The selected PTC-derived cell lines, all sharing *hTERT* promoter mutation, also harbored *BRAF^V600E^* and *TP53* mutations (B-CPAP cell line), *BRAF^V600E^* mutation (K1 cell line) and *RET/PTC* rearrangement (TPC-1 cell line) [13,14]. A human SV40 T large antigen-immortalized thyroid cell line (Nthy-ori 3-1) was used as control. Our results show, for the first time by the analysis of metabolomic profiles, that the pro-oxidant effect of vitamin C in *BRAF* mutant thyroid cancer cells induces the inhibition of glycolysis and alteration of TCA cycle via NAD+ depletion, leading to cell death.

## 2. Materials and Methods

### 2.1. Chemical and Reagents

L-ascorbic acid (vitamin C, catalog number A4544), dimethyl sulfoxide (DMSO), acetonitrile and N-acetyl-L-cysteine (NAC, catalog number A9165) were purchased from Merck (Milan, Italy). Vitamin C and NAC stock solutions (concentration of 0.5 M) were prepared freshly in sterile ultrapure water before each cell treatment.

### 2.2. Cell Culture

PTC-derived TPC-1 and B-CPAP cell lines were kindly provided by Dr. Fusco (Medical School, University Federico II of Naples, Naples, Italy), while the K1 cell line and Nthy-ori3–1, (Simian Virus 40 (SV40)-immortalized normal human thyrocytes) cell lines were purchased from the Health Protection Agency Culture Collections (Health Protection Agency Culture Collections; 2011 http://www.hpa.org.uk, last accessed on 20 May 2021). All cell lines were grown as monolayers in Dulbecco’s Modified Eagle’s Medium/Ham’s F-12 (DMEM/F12) supplemented with 10% fetal bovine serum (FBS, Life Technologies, Milan, Italy), 100 UI/mL penicillin and 100 μg/mL streptomycin (Sigma-Aldrich, Milan, Italy), at 37 °C in a humidified 5% CO_2_ atmosphere.

### 2.3. MTT Viability Test

Vitamin C cytotoxicity was measured using the MTT (3-(4,5-dimethylthiazol-2-yl) -2,5-diphenyltetrazolium bromide) assay as follows. The cells were seeded at the density of 7.5 × 10^3^ cells, in a 96-well plate and incubated for 24 h. Then, vitamin C was added at six different concentrations (0.1–15 mM) and cells were further incubated for 24 h and 48 h, to evaluate the sub-lethal and lethal concentration of vitamin C. After incubation, vitamin C was replaced with 50 μL of MTT reagent (1 mg/mL in DMEM/F12), and cells were incubated for additional 4 h. The resulting formazan crystals were dissolved in 100 μL of DMSO. The absorbances were measured at 570 nm using a TECAN microplate reader (Infinite 200, Tecan, Salzburg, Austria). Viability data were reported as % of control (untreated cells) for each cell line.

### 2.4. Apoptosis Assay

To investigate cell death induced by vitamin C treatment, a flow cytometric analysis using the cell apoptosis kit Annexin V/PropidiumIodide (PI) double staining uptake (Life Technologies, Monza, Italy) was used. Control and PTC-derived cells, at the density of 5 × 10^4^ cells/mL, were seeded in 6-well plates (Corning, Tewksbury, MA, USA) with complete DMEM/F12. First, the cells were treated with 5 mM vitamin C for B-CPAP and K1, 10 mM for TPC-1 and 15 mM for NThy-ori3-1 for 48 h. Next, all cell lines were treated with 5 mM of vitamin C and with 10 mM of NAC, or with NAC and vitamin C together for 48 h. Cells were washed once with PBS 1X and stained, according to the kit’s protocol. Stained cells were then analyzed by flow cytometry, measuring the fluorescence emission at 530 and 620 nm using 488 nm excitation laser (MoFloAstrios EQ, Beckman Coulter). Cell apoptosis was analyzed using Software Summit Version 6.3.1.1, Beckman Coulter. For the following experiments, 5 mM of vitamin C was used in all cell lines.

### 2.5. Determination of Intracellular ROS Production

To detect intracellular ROS production, PTC-derived and control cells were seeded in 96-well plates (7.5 × 10^3^) and grown for 24 h. Cells were then washed with PBS 1X solution and incubated for 30 min with 2′,7′-dichlorofluorescin diacetate probe (H_2_-DCF-DA) (Merck, Milan, Italy) (10 μM), as previously described [15,16]. H_2_-DCF-DA was then removed, and cells were treated with vitamin C (5 mM), with the oxidant tert-butyl hydroperoxide (TBH 2.5 mM), as a positive control, with NAC (10 mM) as an antioxidant, and with combinations of NAC + TBH or vitamin C. After 1 h of incubation, excess of H_2_-DCF-DA was removed and replaced with PBS and then ROS levels were measured by using a microplate reader (Infinite 200, Tecan, Salzburg, Austria) at a controlled temperature of 37 °C. The measurement was performed using an excitation of 490 nm and an emission of 520 nm. ROS production was evaluated for 2 h and monitored taking readings at intervals of 5 min.

### 2.6. Determination of Intracellular Aminothyols

Reduced glutathione (GSH), oxidized glutathione GSSG, cysteine (Cys), and cystine (CySS) levels were determined with a high-performance liquid chromatography coupled with an electrochemical detector (HPLC-ECD), as previously described [17,18,19]. In detail, cells were seeded in 6-well plates at the density of 1 × 10^5^ cells/2 mL and incubated for 24 h. Cells were then treated with vitamin C (5 mM) with or without NAC (10 mM) co-incubation and incubated for 24 h. After the incubation, cells were scraped and extracted with 150 μL of 10% meta-phosphoric acid and 150 μL of 0.05% trifluoroacetic acid (TFA) (Merck, Milan, Italy) solution. After centrifugation, 10 μL of supernatant were collected for the protein determination and the remaining part was injected into the HPLC system. GSH, GSSG, Cys, and CySS amounts were measured using an HPLC (Agilent 1260 infinity, Agilent Technologies, Palo Alto, CA, USA) equipped with an electrochemical detector (DECADE II Antec, Leyden, The Netherlands) and an Agilent interface 35900E. A calibration curve was created using standards of GSH, GSSG, Cys, and CySS (Merck, Milan, Italy), injected with different concentrations. Data were collected and expressed as a ratio between ng of GSH, GSSG, Cys, and CySS, and μg of total proteins.

### 2.7. Aqueous Metabolites Extraction

Cells were seeded in Petri dishes (100 mm) at the density of 1.8 × 10^5^/mL cells and grown for 24 h. Then, cells were treated with 5 mM of vitamin C and further incubated for 24 h. Cells were harvested by scraping with a mixture of cold methanol and water (80:20%) for the polar metabolites extraction as previously described [20]. Cells were then detached and collected in Eppendorf^TM^ tubes. To ensure the complete lysis of the cells, the extraction was combined with 10 min of ultrasonic treatment at a controlled temperature (4 °C). Cell suspensions were centrifuged at 4500 rpm for 30 min at 4 °C. The upper aqueous phase was separated, aliquoted in Eppendorf tubes, and dried in an Eppendorf^TM^ Concentrator Plus overnight.

### 2.8. Glucose Uptake Assay

PTC-derived cells were seeded in 96-well plates (1 × 10^5^ cells/mL) and grown in DMEM/F12 medium for 24 h at 37 °C. Then, cells were rinsed with PBS (pH 7.4) (Euroclone, Pero, ltaly) and treated simultaneously with the fluorescently-tagged glucose derivative 2-[N-(7-nitrobenz-2-oxa-1,3-diazol-4-yl)amino]-2-deoxy-Dglucose (2-NBDG, N13195; ThermoFisher, Waltham, MA, USA) (50 µM) and with vitamin C (5 mM) in PBS for 30 min. Afterwards, external 2-NBDG and the excess of vitamin C were washed off and replaced with 100 µL of fresh PBS. Relative glucose uptake was then measured by reading the fluorescence using a micro plate reader (Infinite 200, Tecan, Salzburg, Austria) at a controlled temperature of 37 °C. The measurements were performed using an excitation wavelength of 485 nm and an emission wavelength of 530 nm.

### 2.9. Ultra High-Performance Liquid Chromatography- Tandem Mass Spectrometry

Aqueous dried samples were quantified using an ultra-performance liquid chromatography coupled with a TSQ Quantiva™ Triple Quadrupole Mass Spectrometer (UHPLC/MS) (ThermoFisher, Waltham, MA, USA)in two targeted analysis as previously described [20]. Briefly, for the first analysis samples were reconstituted with 200 µL of acetonitrile:water (7:3) with ammonium carbonate 0.1 M (Sigma Aldrich, Gillingham, Dorset, UK) and the LC column used was a BEH amide HILIC column (100 × 2.1 mm, 1.7 µm; Waters Ltd., Elstree, Borehamwood, UK). For the second targeted analysis, samples were reconstituted in water with 0,1% of formic acid (Sigma Aldrich, Gillingham, Dorset, UK) and injected using a reverse phase column ACE C18-pfp (150 × 2.1 mm, 2 µm; Advanced Chromatography Technologies. Aberdeen, Scotland). Aqueous metabolites were acquired through selected reaction monitoring (SRM) mass spectrometry analysis using an internal standard mix with: [^13^C, ^15^N] L-proline, L-leucine-d10, L-Valine-d8, L-phenylalanine d5, Succinate-^13^C and Serotonine-d4 (Sigma Aldrich, Gillingham, Dorset, UK), and [^13^C, ^15^N] L-glutamate (Cambridge Isotope Laboratories, Andover, MA, USA) (10 µM each). The Xcalibur software (Thermos fisher scientific, Waltham, MA, USA) was used for data acquisition. Putative recognition of all detected metabolites was performed using a targeted MS/MS analysis and reference to standards for accurate retention times. Peak areas, for each detected metabolite, were then normalized by total area and reported as ranks in the bar graph.

### 2.10. Statistical Analysis

Data were analyzed using the GraphPad Prism v5.0 software (La Jolla, CA, USA). Statistical analysis of the experimental results was performed using unpaired Student *t*-Test. Data are presented as means ± Standard Deviation. All experiments were performed three times independently, each time in triplicate to confirm the results.

## 3. Results

### 3.1. Cytotoxicity Induced by Vitamin C Treatment

Cytotoxicity of vitamin C in PTC-derived cell lines and control cells was analyzed by MTT assay. All cell lines were exposed to different doses of vitamin C (0.5 mM, 1 mM, 2.5 mM, 5 mM, 10 mM, and 15 mM) and cell growth was evaluated after 24 and 48 h. Cell viability was significantly reduced after 48 h of vitamin C (Figure 1), whereas viability was not affected after 24 h incubation (data not shown). B-CPAP and K1 cells were the most sensitive (5 mM), followed by TPC-1 (10 mM). Nthy-ori3-1 cells were the least sensitive to vitamin C (15 mM) (Table 1). Annexin V/Propidium Iodide assay on cells exposed to vitamin C demonstrated that cell death mechanisms involved both apoptosis and necrosis. After 48 h exposure to different concentrations of vitamin C (5 mM for B-CPAP and K1, 10 mM for TPC-1, 15 mM for Nthy-ori3-1), a significant increase of necrosis was observed in all PTC-derived cells (70.05%, 81.32%, and 72.93% for TPC-1, K1, and B-CPAP treated cell lines, respectively). Moreover, a slight increase in apoptosis was induced only in B-CPAP cell line after vitamin C exposure (6.3% untreated vs 20.73% treated cells). On the contrary, in Nthy-ori3-1 cells only apoptosis was detected after vitamin C treatment (Figure 2).

### 3.2. Oxidative Stress Induced by Vitamin C in PTC-Derived Cells

To confirm the pro-oxidant vitamin C effect, changes in the redox balance were assessed by measuring intracellular oxidants and antioxidant species in cells exposed to 5 mM vitamin C (the lowest observed cytotoxic concentration for 2 out of 3 PTC-derived cell lines). *tert*-butyl hydroperoxide (TBH) was used as oxidant positive control and N-acetyl-L-cysteine (NAC) as an antioxidant. Vitamin C-treated B-CPAP cells had significantly higher ROS levels than untreated cells, whereas no difference was observed in K1, TPC-1, and control cells, compared to the corresponding untreated cells. Overall, in all cell lines, NAC was able to reduce ROS levels as expected (Figure 3), and the combined exposure to vitamin C and NAC showed similar ROS level compared to those cells treated only with NAC.

Redox perturbation was further investigated by measuring intracellular antioxidant species. While vitamin C treatment significantly decreased the GSH/GSSG ratio only in B-CPAP cells, Cys/CySS ratio was significantly reduced in K1 cells as well as in B-CPAP cells (Figure 4). Moreover, NAC was able to counteract the vitamin C-induced reduction of GSH/GSSG antioxidant species in B-CPAP cells and decrease in Cys/CySS ratio in K1 cells (Figure 4).

### 3.3. NAC Effect on Vitamin C Induced Cell Death

To investigate the effect of NAC in cytotoxicity induced by vitamin C, PTC-derived and control cell lines were treated with these agents alone or in combination and cell death was assessed by Annexin V/Propidium Iodide assay. As expected, exposure to vitamin C caused cell death by necrosis in 59.7% B-CPAP and 33.9% K1 and by apoptosis in 14.2% B-CPAP and 11.0% K1. Interestingly, exposure to 5 mM vitamin C in combination with NAC (10 mM) caused cell death by necrosis in 8.6% B-CPAP and 4.8% K1, and by apoptosis in 79.7% B-CPAP and 45.2% K1. In TPC-1 and Nthy-ori3-1 control cells, only cell death by apoptosis was observed at a low percentage (<10%) after exposure to vitamin C alone or in combination with NAC (Figure 5).

### 3.4. Alterations in Energetic Metabolism in PTC-Derived Cells Induced by Vitamin C Treatment

To investigate the effect of vitamin C on energy metabolism, a metabolomic-based approach was used to identify metabolic changes in *BRAF* mutant thyroid cancer cells following vitamin C treatment. Since in *BRAF* wild-type thyroid cancer cells (TPC-1) and control cells (Nthy-ori3-1) exposure to 5 mM vitamin C did not affect ROS perturbations and exhibited low sensitivity to cytotoxic effect, the analysis of metabolic profiles was restricted to B-CPAP and K1 cells. Glucose uptake and metabolites involved in glycolysis, TCA cycle, and NAD^+^ salvage pathways were evaluated in these cells after 24 h of vitamin C treatment. Exposure to vitamin C significantly reduced the glucose uptake in PTC-derived cells (Figure 6A) and altered various metabolites in glycolysis and TCA cycle. In particular, in both cell lines, glucose, fructose biphosphate, and glyceraldehyde 3-phosphate were significantly increased in treated cells; dihydroxyacetone phosphate was found at very high levels in K1 cells exposed to vitamin C whereas a slight increase was observed in B-CPAP cells without reaching statistical significance (Figure 6B). On the contrary, glycolysis downstream metabolites (2/3-phosphoglycerate, phosphoenolpyruvate) were decreased in both cell lines but only in K1 were they significantly reduced.

In the TCA cycle, coenzyme A was significantly increased in both vitamin C treated cell lines, whereas acetyl-CoA was significantly reduced only in B-CPAP cells (Figure 6C). The downstream metabolites, such as succinate, fumarate, and malate were significantly increased in vitamin C exposed cells while oxalacetate was found significantly elevated only in K1 cell lines (Figure 6C). Furthermore, metabolites belonging to the NAD^+^ salvage pathway were significantly decreased in treated cancer cells. In particular, vitamin C treatment caused a significant decrease of nicotinamide and NAD^+^ in B-CPAP and K1 (Figure 6D). The summary of the oxidative and metabolic effects of vitamin C is shown in Figure 7. Collectively these results suggest that in thyroid cancer cell lines vitamin C can induce oxidative stress and impairment of glycolysis with reduction of NAD^+^ levels ultimately leading to cell death.

## 4. Discussion

Vitamin C has been gaining growing attention as a potential treatment for human malignancies [21]. It is well known that vitamin C can induce cell death by inducing oxidative stress in cancer cells [22], and there is a growing interest to clarify the cellular mechanisms underlying this effect. Recently, it has been suggested that there is an important relationship between vitamin C pro-oxidant effects and cancer metabolisms in several tumors, such as breast, liver cancers, and leukemia [4,23,24]. This biological link produces dysregulation of several energetic pathways, such as glycolysis, TCA cycle, and pentose phosphate pathway [4,25]. This aspect has not been fully investigated so far in thyroid cancer. In this study, we focused our attention particularly on the cytotoxicity of vitamin C and the modulation of metabolism and redox homeostasis in PTC-derived cell lines harboring different genetic backgrounds [14]. The experiments were carried out on TPC-1 with *RET/PTC1* rearrangement, K1 and B-CPAP harboring *BRAF^V600E^* mutation (in heterozygosis and homozygosis, respectively) [26]. Nthy-ori3.1 cells, derived from human follicular epithelial cells and negative for the above-mentioned alterations, were used as control.

Vitamin C inhibited the growth of PTC and control cells with different susceptibility to the vitamin C treatment (from the less sensitive: Nthy-ori3-1 > TPC-1 > B-CPAP and K1).

It is well known that thyroid hormone synthesis requires hydrogen peroxide (H_2_O_2_) as an oxidative agent. In particular, the process is called “iodide organification”, where specific tyrosine residues of thyroglobulin (TG) undergo iodination oxidative by the enzyme thyroid peroxidase (TPO). The H_2_O_2_ is usually generated by oxidase proteins (DUOX1 and DUOX2) [27]. In a stable human TPO-expressing cell lines, TPO protect DUOX2 from inhibition of H_2_O_2_ by a catalase-like activity, regulating the level of extracellular H_2_O_2_ probably by decreasing the oxidative damage of macromolecules [28]. Buettner’s group demonstrated that a high dose of vitamin C (in the millimolar range) selectively kills cancer cells but not normal cells, because the tumor cells are much less efficient in removing hydrogen peroxide than normal cells. They show that cells with lower amounts of catalase activity were more susceptible to damage and death when they were exposed to high amounts of vitamin C [29]. In our study, Nthy-ori3-1 cells are less susceptible at the higher (15 mM) and lower (5 mM) of the vitamin C concentrations tested. We can suppose that these cells, where TPO is expressed [30], possess a high level of enzymes involve in removing hydrogen peroxide generated from high dose of vitamin C. Therefore, this compound exerted a high cytotoxic activity in PTC cells compared to control cells, even supporting previous studies about the selectively cytotoxic effect of vitamin C in cancer cells harboring *KRAS* or *BRAF* mutations [9,11]. Moreover, it has been demonstrated that aberrant activation of the MAPK pathway related to *BRAF* mutations is crucial in the damage of the iodide-handling machinery [31], so a high cytotoxic activity of vitamin C in B-CPAP and K1 with *BRAF* mutation could be associated with a decrease of the expression of genes involved in thyroid hormone biosynthesis in these cells [32].

Different studies reported that a high concentration of vitamin C induces cell death by apoptosis in various cancer cells [33]. Our results by FACS analysis showed that apoptosis occurred in control cells and necrosis in PTC cells after exposure to vitamin C. Moreover, when the lower concentration of vitamin C (5 mM) was used, a small fraction of TPC-1 and control cells died by apoptosis, while in K1 and B-CPAP, more necrotic cells were detected as compared to the control. Interestingly, the combinatory treatment with vitamin C and NAC lead to a change in the type of cell death of B-CPAP and K1 cell lines shifting from necrosis induced by vitamin C alone to apoptosis induced by exposure to both agents. This observation could be explained by NAC capability to counteract the alteration in redox homeostasis induced by vitamin C. However, the induction of apoptosis can be assumed to be linked to the effects of vitamin C on DNA demethylation. Indeed, vitamin C is known to act as an essential cofactor to numerous monooxygenases and dioxygenases, including the ten-eleven translocation (TET) enzyme family, which catalyzes the hydroxylation of 5-methylcytosine (5mC) to 5-hydroxymethylcytosine (5hmC) in the active process of DNA demethylation [34,35]. This mechanism leads to the reactivation and upregulation of proapoptotic genes, which are conversely downregulated in those cells harboring *KRAS* and *BRAF* mutations, which are in turn known to limit DNA demethylation [36,37,38].

In our experiments, vitamin C increased ROS production only in B-CPAP cells, and this increment was reduced when vitamin C was challenged with NAC. As expected, in B-CPAP the increase in ROS production resulted in a depletion of antioxidants, as demonstrated by the decrease of GSH/GSSG and CyS/CySS ratios. In addition, although ROS production in K1 cells was not significantly higher than untreated cells, the depletion of CyS stocks was also verified, maybe due to higher sensitivity to ROS oxidative action of this cell line. Our data showed that treatments with vitamin C affected mainly the Cys/CySS ratio, suggesting an increased demand for antioxidants, such as GSH, likely due to the high requirement of cysteine for glutathione biosynthesis [39]. Moreover, possibly as a consequence of redox homeostasis unbalance, we observed a metabolic perturbation induced by a high dose of vitamin C in B-CPAP and K1 cells. Indeed, the results showed increased levels of upstream metabolites of glycolysis, suggesting an accumulation of these intermediates inside the cells. This outcome was verified even though glucose uptake was partially inhibited by vitamin C treatments. Indeed, our results showed that exposure to a high-dose of vitamin C resulted in a decrease of relative glucose uptake in PTC-derived and control cells. This could be linked to the fact that vitamin C usually undergoes uptake by cells, before oxidation by dehydroascorbate, through GLUT transporters, due to its chemical structural similarity with glucose [40]. In agreement with this point, recent studies showed in fact that GLUT1 expression is much higher in K1 and B-CPAP cells compared to the control cells [11,41]. The accumulation of glucose and structurally related metabolites may be explained by an impairment of glycolysis, which resulted in decreased downstream metabolites levels. This reduction could be associated with the decrease of NAD^+^ content in PTC-derived cell with *BRAF* mutations, due to vitamin C treatment and its oxidative action as previously well explained by Uetaki et al., [4]. This pathway is fundamental for various cellular processes, including energy metabolism [42]. The depletion of NAD^+^ resulted also in a perturbation of TCA cycle: the upstream metabolites were not significantly regulated, while downstream metabolites were increased. The alteration of glycolysis and TCA cycle is supposed to be responsible for NAD^+^ decrease, confirming what was reported by Uetaki et al., [4]. Furthermore, it has been proposed that high doses of vitamin C may affect mitochondrial homeostasis. In particular, Bakalova and colleagues described the central role of the ascorbyl free radical (AFR) in mediating impairment of mitochondrial respiration [43]. AFR derived from the oxidation of ascorbate and may temporarily accumulate in cells [44,45]. The cytotoxic potential of AFR is strongly correlated with the activity of the NADH:cyto chrome-b5-oxidoreductase-3 (Cyb5R3) which catalyzes rapid conversion of AFR to ascorbate. Cyb5R3 is overexpressed in breast cancer and correlate with poor disease-free and overall survival [46]. Furthermore, increased Cyb5R3 activity is observed in papillary thyroid cancer compared to follicular adenomas [47]. It has been hypothesized that a high intracellular concentration of vitamin C may induce Cyb5R3 activity inhibition, leading to increased level of AFR in the mitochondrial intermembrane space and decreased level of the NAD+/NADH ratio in the cytosol. AFR may also interact with cytochrome C, causing a partial or complete arrest of electron flow in mitochondrial complex and impairing mitochondrial respiration [43]. It could be hypothesized that the same mechanism occurs also in B-CPAP and K1 *BRAF*-mutated cancer cells but this remaining issue requires experimental verification.

## 5. Conclusions

Taken together, our data confirmed that vitamin C can induce ROS production and depletion of antioxidant defenses in PTC cells harboring *BRAF^V600^* mutation but not in cells characterized by *RET/PTC* rearrangements, reinforcing the idea that this compound exerts a selective effect in tumor cells with specific mutations. Moreover, we pointed out that a high concentration of vitamin C has an anti-tumoral effect in PTC cells, by altering redox homeostasis which had an impact on the NAD salvage pathway, resulting in turn in glycolysis and TCA cycle impairment, and finally inducing cell death. Howsoever, further investigations, including in vivo studies, are needed to better elucidate the mechanisms involved in the cytotoxic action of vitamin C in PTC-derived cells, to promote its potential use as anti-tumoral compound.

## Figures and Tables

**Figure 1 antioxidants-10-00809-f001:**
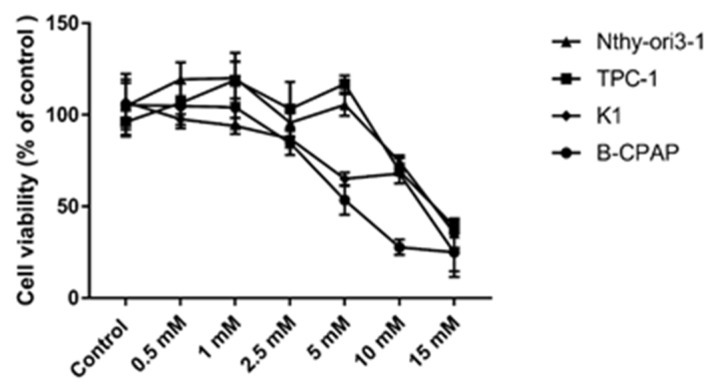
Viability of cells treated with vitamin C (0.5–15 mM) for 48 h analyzed by MTT assay. Data are expressed as % of the untreated cells (control) ± SD. All experiments were performed three times independently.

**Figure 2 antioxidants-10-00809-f002:**
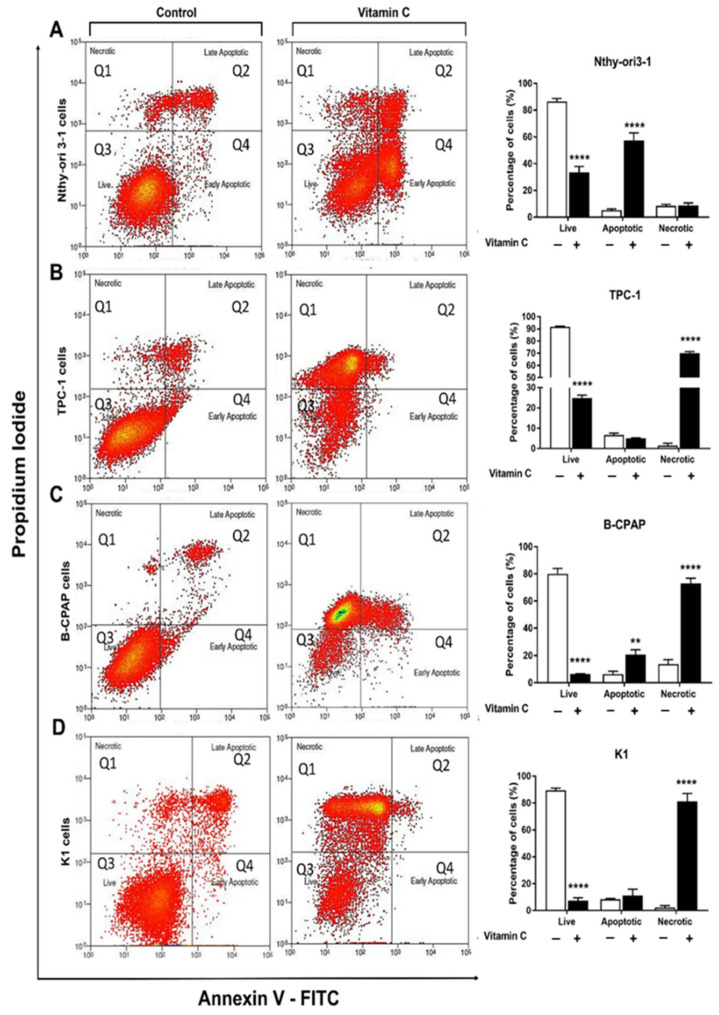
Percentage of live, apoptotic, and necrotic cells measured by flow cytometry using the PI-annexin V assay. Dot plots showing cell death in all cell lines treated for 48h with different doses of vitamin C: Nthy-ori 3-1 with 15 mM (**A**), TPC-1 with 10 mM (**B**), K1 and B-CPAP with 5 mM (**C**,**D**). Bar graphs are representative of three experiments and show the percentage of live, apoptotic (early apoptotic and late apoptotic cells), and necrotic cells. Data are expressed as the Mean of % cells ± SD. All experiments were performed three times independently, each time in triplicate, to confirm the results. Statistical analyses were performed by Student *t*-test. Statistical differences vs. control are expressed with superscript symbols: ** *p* < 0.01; **** *p* < 0.0001.

**Figure 3 antioxidants-10-00809-f003:**
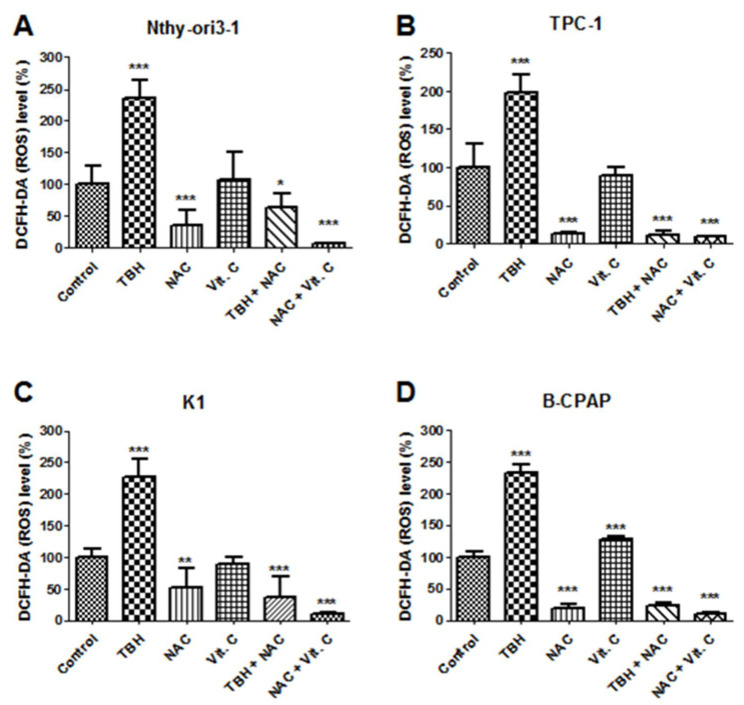
Reactive oxygen species (ROS) levels in PTC-derived cell lines and non-tumoral thyroid cell line, expressed as % of fluorescence of control (untreated cells for each cell line) after treatment with vitamin C 5 mM, TBH 2.5 mM, NAC 10 mM and a combination of TBH+NAC and NAC + vitamin C. (**A**) Nthy-ori3-1, (**B**) TPC-1 (**C**) K1 (**D**) B-CPAP cell lines. Statistical analysis was performed by Student *t*-Test. Results were considered significant vs. control when * *p* < 0.05, ** *p* < 0.01, and *** *p* < 0.001.

**Figure 4 antioxidants-10-00809-f004:**
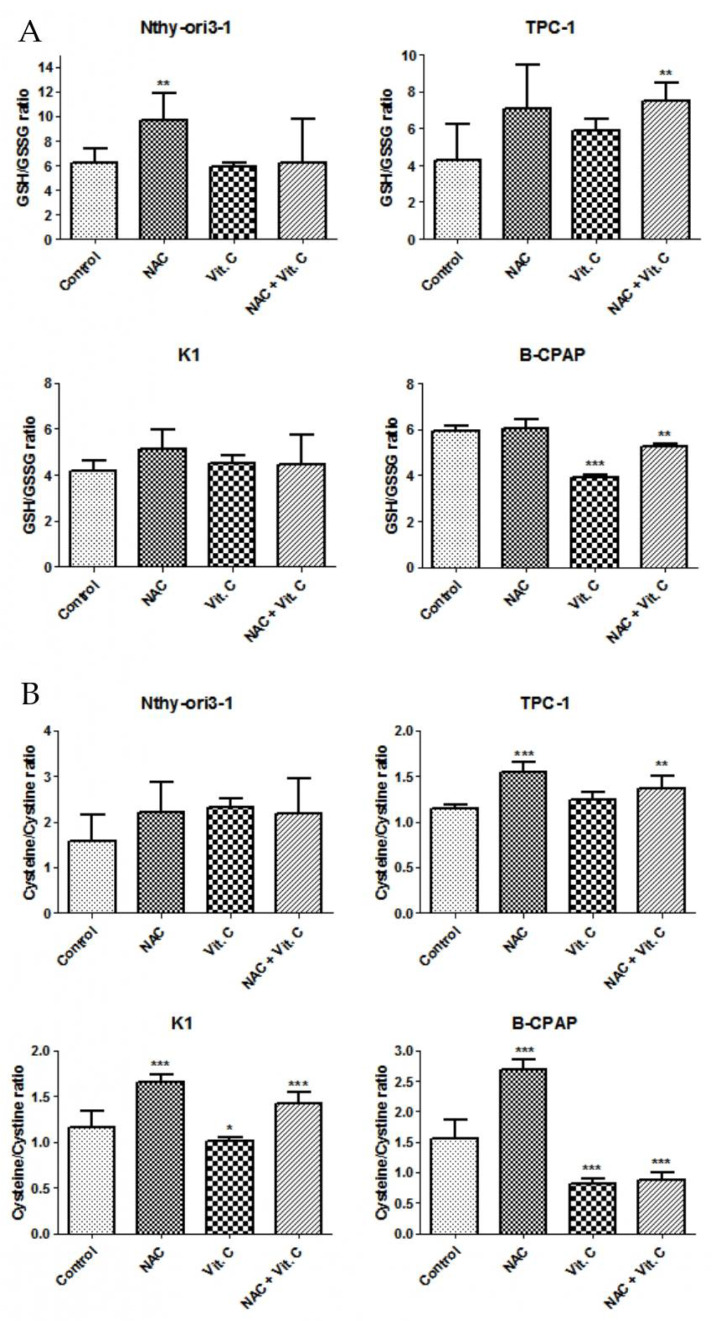
Levels of GSH/GSSG ratio (**A**) in PTC-derived cell lines and control cell line and levels of Cys/CySS ratio (**B**) after 24 h of incubation with vitamin C alone (5 mM) or in combination with NAC (10 mM). Peak areas of intracellular aminothiols were normalized using protein contents (ng of aminothiols per µg of proteins) and expressed as the ratio of reduced and oxidized forms. Data are expressed as Mean ± SD. All experiments were performed three times independently, each time in triplicate to confirm the results. Statistical analysis was performed by Student *t*-test. Results were considered significant vs. control when * *p* < 0.05, ** *p* < 0.01, and *** *p* < 0.001.

**Figure 5 antioxidants-10-00809-f005:**
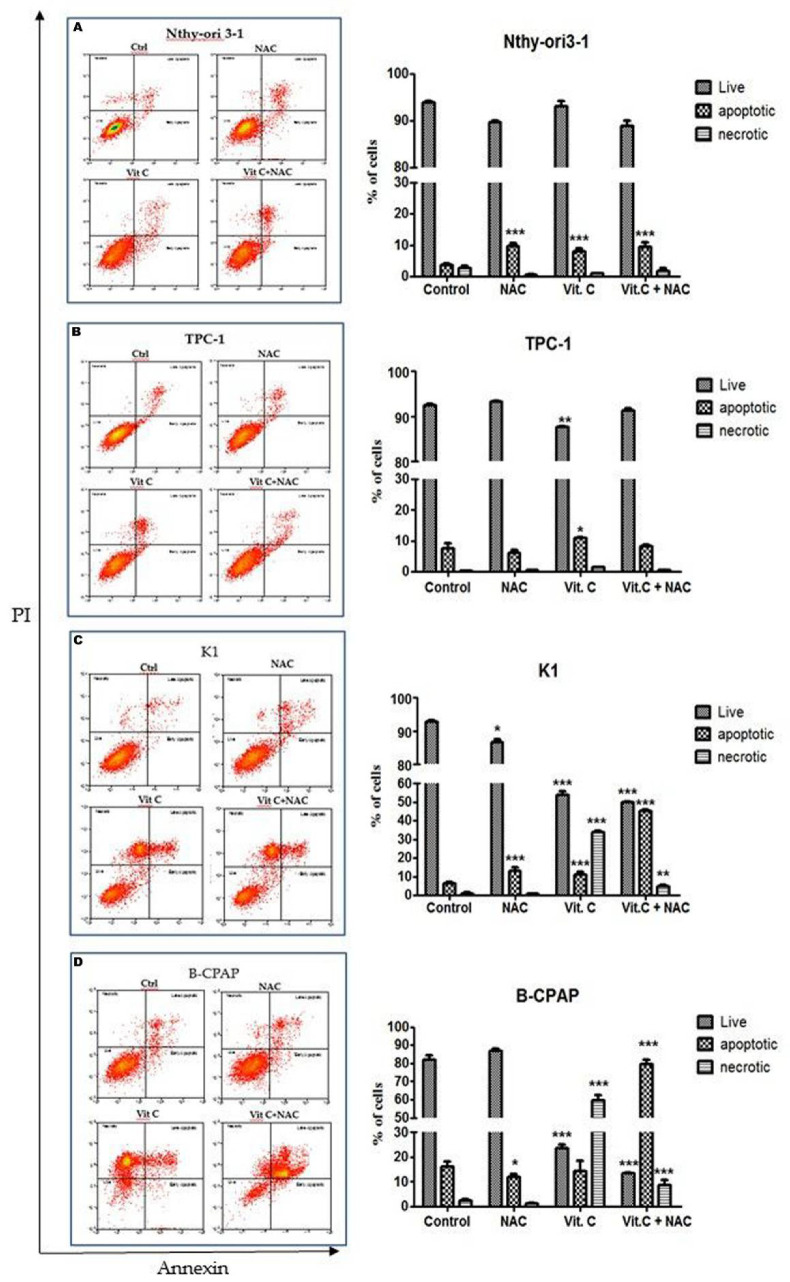
Percentage of live, apoptotic, and necrotic cells measured by flow cytometry using the PI-annexin V assay. All cell lines were exposed to 5 mM of vitamin C or NAC (10 mM) or a combination of both agents for 48 h. Dot plots showing cell death in Nthy-ori3-1 (**A**), TPC-1 (**B**), K1 (**C**), and B-CPAP cells (**D**). Pictures are representative of three experiments. Bar graphs represented the percentage of live, apoptotic (early apoptotic and late apoptotic cells), and necrotic cells. Data are expressed as Mean of % cells ± SD. All experiments were performed three times independently, each time in triplicate, to confirm the results. Statistical analyses were performed by Student *t*-test. Statistical differences vs. control are expressed with superscript symbols: * *p* < 0.05; ** *p* < 0.01; *** *p* < 0.001.

**Figure 6 antioxidants-10-00809-f006:**
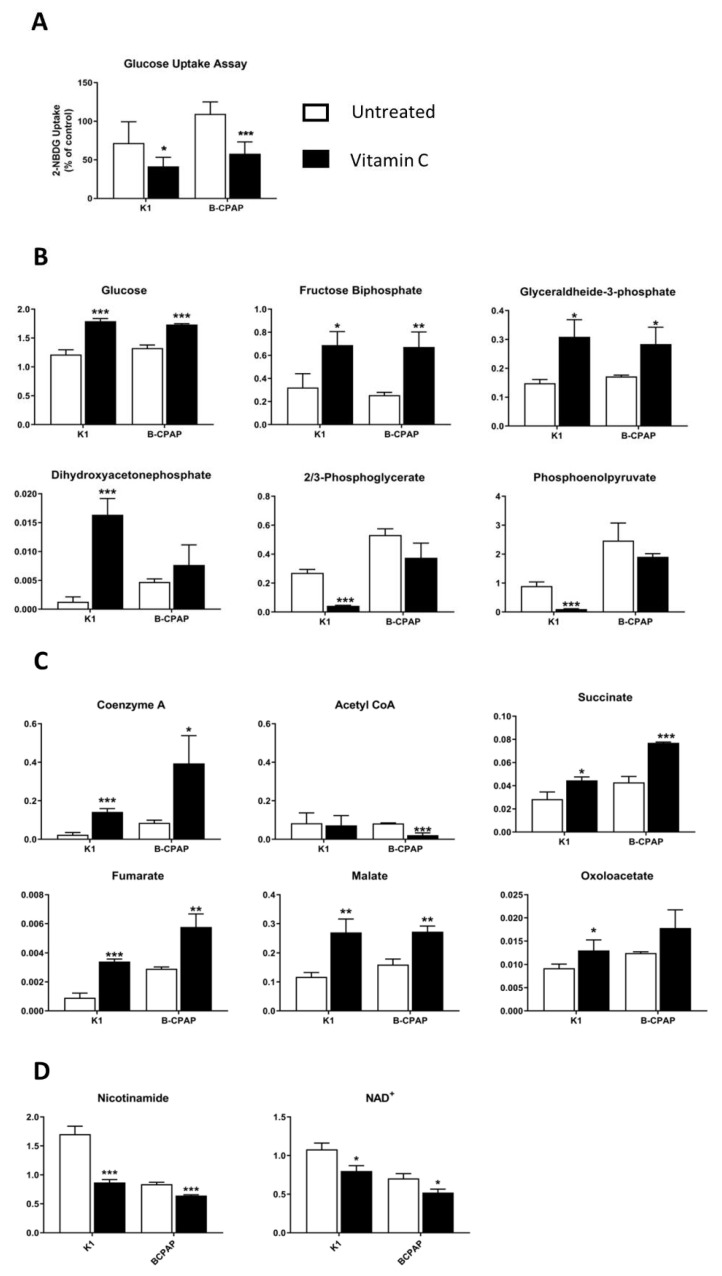
Metabolic alterations induced by vitamin C. (**A**) Quantification of the relative glucose uptake was done through the fluorescent glucose analog 2-NBDG in cancer cells treated with vitamin C. Data are expressed as media ± SD. (**B**) Intracellular levels of glycolytic pathway; (**C**) TCA cycle; (**D**) NAD^+^ salvage pathway, and ATP level. Metabolites were measured after 24 h of incubation with vitamin C. Bar graphs indicate the peak areas of the metabolites, normalized for total area, and expressed in the graphs as ranks. All experiments were performed three times independently, each time in triplicate, to confirm the results. Statistical analyses were performed by Student *t*-test. Results were considered significant when: * *p* < 0.05; ** *p* < 0.01; *** *p* < 0.001.

**Figure 7 antioxidants-10-00809-f007:**
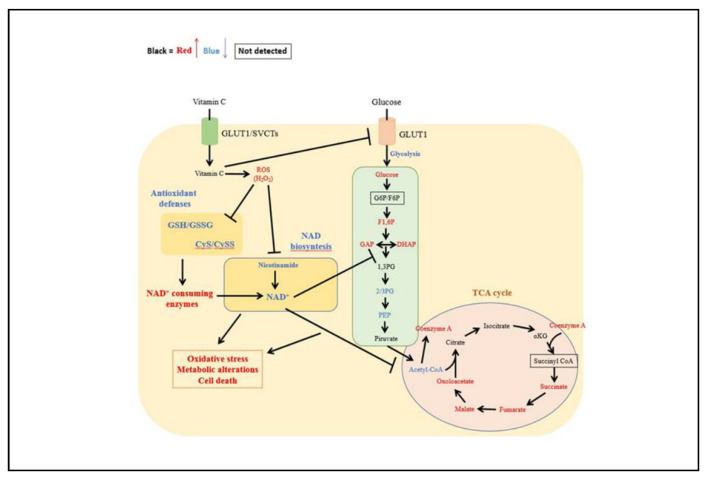
Summary of the effect of a high dose of vitamin C in B-CPAP and K1 cells. Vitamin C induces oxidative stress by increasing ROS and therefore decreasing antioxidants. The decrease of GSH and CyS results in deregulation of the NAD^+^ biosynthesis and consequently in an impairment of energetic pathways, such as glycolysis and TCA cycle. Color coding indicates significantly different metabolites (red, upregulated; blue, downregulated; black, unchanged; boxed, not detected).

**Table 1 antioxidants-10-00809-t001:** Concentration of vitamin C able to induce 50% of cell death in PTC-derived and control cells.

Cell Lines	Vitamin C Concentration
B-CPAP	5 mM
K1	5 mM
TPC-1	10 mM
Nthy-ori3-1	15 mM

## Data Availability

The data presented in this study are showed in this paper.

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
