# Peer review of "Vitamin C Cytotoxicity and Its Effects in Redox Homeostasis and Energetic Metabolism in Papillary Thyroid Carcinoma Cell Lines"

_antioxidants, 2021, doi:10.3390/antiox10050809_

Round 1

Reviewer 1 Report

The manuscript by Tronci et al investigates the effects of Vitamin C on energetic metabolism in papillary thyroid carcinoma cell lines. The manuscript is well written and well organised. The main conclusion are properly supported by the data. Nonetheless, the novelty of the overall study is not clear compare to the state-of-the-art in the field.

Major :

Figure 6A : it is unclear to me why not all cell lines are used in this experiments. It might be interesting to include all cell lines to reinforce the conclusion in this figure and analysis.

Lane 429 in discussion : there is mention of different mutation in different cell lines. This is not indicated earlier in the manuscript, neither is it clear which cell line harbor which mutation.

DNA methylation is impacted by vitamin C. It is thus unclear how it might interfere with the  results presented in this data.

Minor point :

  • Please clarify why « ascorbate » is used instead of « vitamin C » in some sentences in the text.
  • Please do not use «  = » when indicating the P values in the legends.

Author Response

Reviewer #1.

The manuscript by Tronci et al investigates the effects of Vitamin C on energetic metabolism in papillary thyroid carcinoma cell lines. The manuscript is well written and well organised. The main conclusion are properly supported by the data. Nonetheless, the novelty of the overall study is not clear compare to the state-of-the-art in the field.

We do appreciate the Reviewer’s comment and suggestion. It is well known that in several kinds of solid tumor such as colon, breast, prostate, and bladder cancer (Yun et al., 2015; Soo Jung Lee et al., 2019; Maramag et al.;1997; Peng et al., 2018), vitamin C induces ROS production leading to cell death via distinct mechanisms. The effect of a high dose of vitamin C in papillary thyroid cancer is poorly investigated, in particular for its effect on thyroid cancer cells metabolism. In 2019, Xi et al demonstrated that vitamin C inhibited thyroid cancer cell proliferation in BRAF wild-type thyroid cancer cells by inducing ROS, leading to a decrease in the activity of EGF/EGFR-MAPK/ERK signaling and an increase in AKT ubiquitination and degradation. In BRAF mutant thyroid cancer cells, vitamin C induced cell death by inhibiting the activity of ATP-dependent MAPK/ERK signaling and by inducing proteasome degradation of AKT via the ROS-dependent pathway. Recently, the same group (Xi et al., 2021) reinforced the idea of the antitumor effect of vitamin C in a combination with a PLX4032 (a selective oral inhibitor of the BRAFV600E kinase) in BRAF mutant thyroid cancer cells. In our study, we confirmed the pro-oxidant effect of vitamin C in BRAF mutant thyroid cancer cells as relevant in triggering the cytotoxicity in PTC cells. Analysis of metabolomic profiles also suggests that inhibition of glycolysis and alteration of TCA cycle via NAD+ depletion can play an important role in this mechanism of PTC cancer cell death. Following the Reviewer’s suggestions, now we have stressed the novelty of our work in the Introduction section (Page 2, lines 58 -61).

Major :

Figure 6A : it is unclear to me why not all cell lines are used in this experiments. It might be interesting to include all cell lines to reinforce the conclusion in this figure and analysis.

Thanks to the reviewer. In this study we have firstly determined the concentration of vitamin C able to induce 50% of cell death in PTC-derived and control cells by MTT assay. The results confirm that PTC cells harboring BRAF mutation (B-CPAP and K1 cells) were more sensitive than PTC cells with RET/PTC rearrangement (TPC-1) and control cells (Nthy-ori3-1). On this basis, we decided to use the lowest cytotoxic concentration (5 mM) to perform the subsequent experiments. The assessment of ROS production and intracellular antioxidant species demonstrated that 5 mM vitamin C induce ROS perturbation only in PTC cells harboring BRAF mutation leading to cell death. Since TPC-1 and Nthy-ori3-1 cells exposure to 5 mM vitamin C did not affect ROS production and intracellular antioxidant species levels, and exhibited low sensitivity to cytotoxic effect, we decided to perform the analysis of the metabolic profile only in B-CPAP and K1 cells. For greater clarity, we have better explained in the Results section why we did not extended the metabolomic profiling to all the cell lines used in this investigation (Page 11, lines 296 – 301).

Lane 429 in discussion: there is mention of different mutation in different cell lines. This is not indicated earlier in the manuscript, neither is it clear which cell line harbor which mutation.

We agree with this comment, and we have added more information about PTC-cell lines in the Discussion section. Please see more details in Page 13, lines 385-389.

DNA methylation is impacted by vitamin C. It is thus unclear how it might interfere with the results presented in this data.

Thanks to the Reviewer for this interesting input to better clarify the vitamin C anti-cancer mechanism in PTC cells. It is well known that DNA methylation is somewhat limited by vitamin C which vice versa has been shown to induce DNA demethylation. This demethylation induces the activation of proapoptotic genes which is reflected in the increase of apoptosis. This was observed above all in K1 and B-CPAP cells when treated also with the antioxidant NAC, where vitamin C acted by activating apoptosis and, to a lesser extent, necrosis. Necrosis instead occurred in cells treated with only vitamin C, which induced unsustainable oxidative stress. Thanks to this suggestion we have discussed in detail this aspect (Page 14, lines 425 - 433) to better enucleate the function of vitamin C in the context of this study.

Minor point :

  • Please clarify why « ascorbate » is used instead of « vitamin C » in some sentences in the text.

We used ascorbate as a synonym to avoid “vitamin C” repetitions in the manuscript sentences. For clarity, we have left only “vitamin C” and removed ascorbate along the text.

  • Please do not use «  = » when indicating the P values in the legends.

The symbol ‘=’ has been removed from all figure legends.

Reviewer 2 Report

Tronci and co-workers studied the effect of vitamin C on cytotoxicity, redox homeostasis and energetic metabolism in papillary thyroid cancer-derived cell lines (TPC-1, B-CPAP and K1). Immortalized human thyrocytes (Nthy-ori3-1) were used as a control. The manuscript is interesting, however, it cannot be accepted in the current form. Suggestions to enable publication of a revised manuscript are listed below.

  1. The Authors should mention about oxidase stress in normal thyrocytes. For example, DUOXs (DUOX1 and DUOX2) actively synthetize hydrogen peroxide (H2O2) required for iodide organification during thyroid peroxidase (TPO)-mediated hormonogenesis. It has been already demonstrated, that Nthy-ori3-1 cells express surface-exposed enzymatically active TPO (doi: https://doi.org/10.1371/journal.pone.0193624). Moreover, Fortunato et al. (doi: https://doi.org/10.1210/jc.2010-1085) demonstrated that TPO presents a catalase-like effect.
  2. The second important issue is the down-regulation of genes involved in thyroid hormone biosynthesis in thyroid cancer cells in comparison with normal thyrocytes. The cause of this phenomenon is the activation of MAP kinase (MAPK) signaling pathway, especially due to BRAFV600E point mutation.
  3. Please provide in detail how the vitamin C solution was prepared (stock concentration, diluent, catalog number, storage condition, etc.). Was the stock frishly prepared each time?
  4. Please provide pH value of PBS in line 139 on page 4.
  5. Please rephrase section 2.7. (Glucose uptake assay).
  6. Please complete the information on origin for each reagent/equipment (city, country; eg., in lines 141, 153, 156)

Typos/language mistakes (examples):

  1. Lines 118-119: Please rephrase „an aliquot of 10 μL of supernatant was collected”;
  2. Line 263: „BCPAP” should be changed with „B-CPAP”;
  3. Lines 267 and 268: „(5mM)” and „(10mM)” should be changed with „(5 mM)” and „(10 mM)”, respectively;
  4. Line 285: „5mM of Vitamin C” should be changed with „5 mM of vitamin C”.

Author Response

Reviewer #2

Tronci and co-workers studied the effect of vitamin C on cytotoxicity, redox homeostasis and energetic metabolism in papillary thyroid cancer-derived cell lines (TPC-1, B-CPAP and K1). Immortalized human thyrocytes (Nthy-ori3-1) were used as a control. The manuscript is interesting, however, it cannot be accepted in the current form. Suggestions to enable publication of a revised manuscript are listed below.

The Authors should mention about oxidase stress in normal thyrocytes. For example, DUOXs (DUOX1 and DUOX2) actively synthetize hydrogen peroxide (H2O2) required for iodide organification during thyroid peroxidase (TPO)-mediated hormonogenesis. It has been already demonstrated, that Nthy-ori3-1 cells express surface-exposed enzymatically active TPO (doi: https://doi.org/10.1371/journal.pone.0193624). Moreover, Fortunato et al. (doi: https://doi.org/10.1210/jc.2010-1085) demonstrated that TPO presents a catalase-like effect.

We do appreciate the Reviewer ‘s comments and suggestion. We have added comments about the importance of redox homeostasis, regulated by TPO and oxidase proteins, in thyroid hormonogenesis. Moreover, we have evaluated the possible link between the effect of vitamin C obtained in control cells and the role of enzymes involved in the regulation of H2O2 production in thyroid hormone synthesis. Please see more details in the Discussion section in Page 14 and lines 393 – 408.

The second important issue is the down-regulation of genes involved in thyroid hormone biosynthesis in thyroid cancer cells in comparison with normal thyrocytes. The cause of this phenomenon is the activation of MAP kinase (MAPK) signaling pathway, especially due to BRAFV600E point mutation.

We do appreciate the Reviewer’s suggestion. We have added the comments in the Discussion section (Page 14, lines 410 – 415).

Please provide in detail how the vitamin C solution was prepared (stock concentration, diluent, catalog number, storage condition, etc.). Was the stock frishly prepared each time?

All the required details have been reported in the new section 2.1. Chemical and reagents (Page 2, lines 63 – 67).

Please provide pH value of PBS in line 139 on page 4.

The value has been added as suggested.

Please rephrase section 2.7. (Glucose uptake assay).

The section (now 2.8.) has been rephrased as suggested to better explain the procedure which was followed for the assessment of glucose uptake.

Please complete the information on origin for each reagent/equipment (city, country; eg., in lines 141, 153, 156)

The details on reagents and equipments have been updated as requested.

Typos/language mistakes (examples):

Lines 118-119: Please rephrase „an aliquot of 10 μL of supernatant was collected”;

The sentence has been rephrased.

Line 263: „BCPAP” should be changed with „B-CPAP”;

Changed.

Lines 267 and 268: „(5mM)” and „(10mM)” should be changed with „(5 mM)” and „(10 mM)”, respectively;

Changed.

Line 285: „5mM of Vitamin C” should be changed with „5 mM of vitamin C”

Changed.

Reviewer 3 Report

The manuscript mentioned the possible mechanisms of VC on PTC cancer cell death. The manuscript is full of important information for understanding the anti-tumor mechanism of VC. I can recommend acceptance of the manuscript in the journal of “antioxidants”, however, the manuscript would be further improved if the authors perform additional experiment to evaluate mitochondrial respiration, as written below. - The readers of this manuscript would want to know how mitochondrial activity is altered by the incubation of high concentration of VC. The commonly recognized theory of cellular respiration is that the decrease of glycolysis leads to the increase of mitochondrial activity. However, the alteration of mitochondrial respiration is hard to understand by Figure 7. Since the amount of ATP production does not always correlate with mitochondrial activity, the measurement OCR (oxygen consumption ratio) would be recommended. If VC decreases not only glycolysis but also mitochondrial respiration, that could be one of the reasons why VC elicits tumor cell death. In case the additional experiment is impossible by the authors, they should add some comments regarding VC- induced alteration of mitochondrial respiration at least in the Discussion.

Author Response

Reviewer #3

The manuscript mentioned the possible mechanisms of VC on PTC cancer cell death. The manuscript is full of important information for understanding the anti-tumor mechanism of VC. I can recommend acceptance of the manuscript in the journal of “antioxidants”, however, the manuscript would be further improved if the authors perform additional experiment to evaluate mitochondrial respiration, as written below. - The readers of this manuscript would want to know how mitochondrial activity is altered by the incubation of high concentration of VC. The commonly recognized theory of cellular respiration is that the decrease of glycolysis leads to the increase of mitochondrial activity. However, the alteration of mitochondrial respiration is hard to understand by Figure 7. Since the amount of ATP production does not always correlate with mitochondrial activity, the measurement OCR (oxygen consumption ratio) would be recommended. If VC decreases not only glycolysis but also mitochondrial respiration, that could be one of the reasons why VC elicits tumor cell death. In case the additional experiment is impossible by the authors, they should add some comments regarding VC- induced alteration of mitochondrial respiration at least in the Discussion.

We thank the Reviewer for the interesting suggestion. Unfortunately, it will not be possible to carry out experiments of this type quickly enough to let us insert the results in this manuscript. The measurement of oxygen consumption ratio (OCR) could represent a very important step which we will then consider for further future investigation. In particular, it would be interesting to evaluate the effects of vitamin C on mitochondrial activity in cancer stem-like cells isolated from PTC-derived cell lines using 3D cultures (tyrosphere) and in parental cells (2D cultures), given the results already reported by our research group (Metabolomic alterations in thyrospheres and adherent parental cells in papillary thyroid carcinoma cell lines: A pilot study. International Journal of Molecular Sciences, 2018, 19(10), 2948, doi 10.3390/ijms19102948).

However, based on the literature evidence, mitochondrial dysfunction is an important target for vitamin C and represents a typical hallmark of cancer. Several mechanisms were reported to elucidate the interaction between vitamin C and mitochondria. In particular, Bakalova and colleagues described the central role of the ascorbyl free radical (AFR) in mediating impairment of mitochondrial respiration. AFR derived from the oxidation of ascorbate and may temporarily accumulate in cells. The cytotoxic potential of AFR is strongly correlated with the activity of the NADH:cytochrome-b5-oxidoreductase-3 (Cyb5R3) which catalyzes the rapid conversion of AFR to ascorbate. Cyb5R3 is overexpressed in breast cancer and correlates with poor disease-free and overall survival (Lund et al.2015). Furthermore, increased Cyb5R3 activity is observed in papillary thyroid cancer compared to follicular adenomas (Proskurnina et al. 2020). It has been hypothesized that a high intracellular concentration of vitamin C may induce Cyb5R3 activity inhibition, leading to increased levels of AFR in the mitochondrial intermembrane space and  decrease levels of the NAD+/NADH ratio in the cytosol. AFR may also interact with cytochrome C, causing a partial or complete arrest of electron flow in mitochondrial complex and impairing mitochondrial respiration (Bakalova et al. 2020). It could be hypothesized that the same mechanism occurs also in B-CPAP and K1 BRAF mutated cancer cells but this remaining issue requires experimental verification. We have added these comments in the Discussion section (Page 15, lines 463 - 479).

Round 2

Reviewer 2 Report

I recommend the manuscript to be published in its current form.

Reviewer 3 Report

Although this reviewer expected the data that shows VC-induced alteration of oxygen consumption in TCA cycle, the authors supplemented the explanation that deserves the acceptance of the manuscript.